# Milling Stability Prediction: A New Approach Based on a Composited Newton–Cotes Formula

**DOI:** 10.3390/mi14071304

**Published:** 2023-06-25

**Authors:** Junqiang Zheng, Pengfei Ren, Chaofeng Zhou, Xu Du

**Affiliations:** 1School of Mechanical Engineering, Hefei University of Technology, Hefei 230009, China; zhengjunqiang@hdu.edu.cn; 2School of Mechanical Engineering, Hangzhou Dianzi University, Hangzhou 310018, China; 3School of Mechanical Engineering, Zhejiang Sci-Tech University, Hangzhou 310018, China; 202130605261@mails.zstu.edu.cn (P.R.); 202230503321@mails.zstu.edu.cn (C.Z.)

**Keywords:** dynamic model, regenerative chatter, milling stability prediction, composite cotes formula, Floquet theory

## Abstract

Based on a composited Newton–Cotes formula, this paper proposes a numerical method to predict milling stability considering regenerative chatter and focusing on rate and prediction accuracy. First, the dynamic model of milling motion is expressed as state-space equations considering regenerative chatter, with the tooth passing period divided into a set of time intervals. Second, a composited Newton–Cotes formula is introduced to calculate the transition function map for each time interval. Third, the state transition matrix is constructed based on the above-mentioned transition function, and the prediction stability boundary is determined by the Floquet theory. Finally, simulation analysis and experimental verification are conducted to verify the effectiveness of the proposed method. The simulation results demonstrate that, for the milling model with a single degree of freedom (DOF), the convergence rate and prediction accuracy of the proposed method are higher than those of the comparison method. The experimental results demonstrate that, for the milling model with two DOFs, the machining parameters below the prediction stability boundary can avoid the chatter as much as possible, ensuring the machined surface quality.

## 1. Introduction

Regenerative chatter arising from a self-excitation chatter vibration may be unavoidable in milling processes, which will reduce machining quality and efficiency and accelerate tool wear [1,2]. Researchers have attempted to avoid regenerative chatter using prediction [3,4], identification [5,6,7], and suppression [8,9,10] methods. The research results demonstrate that selecting machining parameters to avoid chatter occurrence is vital and that the stability lobe diagrams provide a reference for appropriately selecting machining parameters [11]. The objective of this study is to develop a method to draw stability lobe diagrams with high precision. 

As regenerative chatter is integrated into the dynamics model, the dynamics model is reformulated as delay differential equations (DDEs) with time-periodic coefficients [12,13]. So far, many methods, including experimental, analytical, and numerical methods, have been proposed to solve the DDEs to obtain stability lobe diagrams. Based on the zeroth-order harmonics of the Fourier series expansion, Altintas et al. [14] transformed the DDEs into the Laplace frequency, which is more suitable for high-radical-immersion milling. Taking higher-order harmonics into account, Merdol et al. [15] proposed a multi-frequency solution for low-radical-immersion milling, which is also called the multi-frequency method. In addition to the above-mentioned stability analysis methods in the frequency domain, there are some stability analysis methods in the time domain. Insperger et al. [16,17] first presented a semi-discretization method (SDM) for a delayed system and then presented an updated version of the SDM (first-order SDM) for periodic systems with a single discrete time delay. By extending the SDM and first-order SDM, the second-order SDM (second-order SDM) has also been proposed for variable spindle speed milling [18]. Ding et al. [19] proposed a full-discretization method (FDM) in virtue of a direct integration scheme. To further improve the prediction accuracy, Ding et al. [20] and Yan et al. [21] presented the second-order FDM and third-order FDM, respectively. Based on the integral equation and numerical integration formulas, Ding et al. [22] developed a concise semi-analytical method for constructing a discrete dynamical map approximating the DDEs. Dong et al. [23] have recently proposed an updated numerical integration method. Li et al. [24] proposed an accurate and fast milling stability prediction approach based on the Newton–Cotes method (NCM). In addition to the above-mentioned methods, other methods such as the Adams–Moulton-based method [25], the spectral element method [26], and the Lyapunov–Krasovskii method [27] exist. The aforementioned stability analysis methods mainly focus on three targets: convergence rate, prediction accuracy, and computational efficiency. These three targets usually contradict each other to a certain extent. Based on a composited Newton–Cotes formula, this paper presents a novel milling stability prediction method, i.e., the composited Newton–Cotes method (CNCM), with a high convergence rate and prediction accuracy.

The remainder of this paper is organized as follows. Section 2 provides a brief description of the composited Newton–Cotes formula. The composited Newton–Cotes-based milling stability analysis method is presented in Section 3. Comparisons of the convergence rate, prediction accuracy, and computational efficiency between the proposed method and NCM are presented in Section 4, and experimental verification is presented in Section 5. Finally, conclusions are drawn in Section 6.

## 2. Composited Newton–Cotes Formula

The nth-order Newton–Cotes formula has at least n-order algebraic accuracy [28], which is defined as
(1)$$\int_{a}^{b}f\left(t\right)dt=\left(b-a\right)\sum_{k=0}^{n}C_{k}^{\left(n\right)}f\left(x_{k}\right)$$
where [a, b] denotes the parameter interval, and it is divided into n equal intervals. The span of each interval is defined as $\left(b-a\right)/n$, and then $x_{k}=a+k\left(b-a\right)/n$ is deduced. The $C_{k}^{\left(n\right)}$ denotes the cotes coefficient and is defined as
(2)$$C_{k}^{\left(n\right)}=\frac{{\left(-1\right)}^{n-k}}{nk!\left(n-k\right)}\int_{0}^{n}\overset{n}{\underset{j=0,j\neq k}{\prod }}\left(\tau -j\right)d\tau \;k=0,1,\cdots ,n,$$
where depends only on n. It is worth mentioning that the approximation accuracy of Equation (1) is significantly reduced when n is greater than or equal to eight. Increasing n cannot ensure the approximation accuracy of Equation (1). To this end, the composited Newton–Cotes formula is introduced below. To improve the integration accuracy, mid-points of the interval should be considered. So the interval $\left[x_{k},x_{k+1}\right]$ is divided into four equal sub-intervals: $x_{k}$, $x_{k+1/4}$, $x_{k+1/2}$, $x_{k+3/4}$, and $x_{k+1}$. The lower-order Newton–Cotes formula is used in each sub-interval, and then the approximation results of each subinterval are finally added together. According to Bool’s rule (*n* = 4), the fourth-order composited Newton–Cotes formula [29] can be introduced as follows:(3)$$\begin{array}{ll} \int_{a}^{b}f\left(t\right)dt= & \frac{7\Delta }{90}\left[f\left(a\right)+\sum_{k=0}^{n-1}f\left(x_{k}\right)+\sum_{k=0}^{n-1}f\left(x_{k+1}\right)+f\left(b\right)\right]\\ & +\frac{16\Delta }{45}\left[\sum_{k=0}^{n-1}f\left(x_{k+1/4}\right)+\sum_{k=0}^{n-1}f\left(x_{k+3/4}\right)\right]+\frac{2\Delta }{15}\sum_{k=0}^{n-1}f\left(x_{k+1/2}\right) \end{array}$$
where $\Delta =\left(b-a\right)/n$. The remainder of the equation above is
(4)$$R_{n}=-\frac{2\left(b-a\right)}{945}{\left(\frac{\Delta }{4}\right)}^{6}f^{6}\left(u\right)\;u\in \left[a,b\right].$$

## 3. Mathematical Model

In this work, it is assumed that the workpiece is rigid, that the milling cutters have two DOFs and that there is no tool run-out situation in the milling process. The tool helix angle is not considered either. Considering the regenerative chatter, the mathematical model of milling motion can be formulated as a second-order differential Equation (24) as follows:(5)$$M\ddot{q}\left(t\right)+C\dot{q}\left(t\right)+Kq\left(t\right)=-a_{p}K_{C}\left(T\right)\left[q\left(t\right)-q\left(t-T\right)\right],$$
where **M**, **C**, and **K** denote the modal stiffness, mass, and damping matrices of the milling system, respectively; $q\left(t\right)$ denotes the displacement vector, $\dot{q}\left(t\right)$ and $\ddot{q}\left(t\right)$ denote its first- and second-order derivatives with respect to t; $a_{p}$ denotes the depth of cut; $K_{C}\left(t\right)$ is a time-periodic coefficient matrix that satisfies $K_{C}\left(t\right)=K_{C}\left(t+T\right)$; T denotes the time delay. For the milling cutter, it is the same as one-tooth passing period, i.e., $T=60/\left(N\Omega \right)$, where N denotes the number of tool teeth and Ω denotes the spindle speed (rpm).

A new vector $x\left(t\right)$ as $x\left(t\right)={\left[q\left(t\right),\dot{q}\left(t\right)\right]}^{T}$ is defined, and then the state-space expression of (5), can be expressed as [22]
(6)$$\dot{x}\left(t\right)=Ax\left(t\right)+a_{p}B\left(t\right)\left[x\left(t\right)-x\left(t-T\right)\right]$$
where
(7)$$A=\left(\begin{array}{cc} 0 & I\\ -M^{-1}K & -M^{-1}C \end{array}\right)$$
(8)$$B\left(t\right)=\left(\begin{array}{cc} 0 & 0\\ -M^{-1}K_{C}\left(t\right) & 0 \end{array}\right)$$
(9)$$K_{C}\left(t\right)=\left(\begin{array}{cc} h_{xx}\left(t\right) & 0\\ 0 & 0 \end{array}\right)$$
(10)$$h_{xx}\left(t\right)=\sum_{j=1}^{N}g\left(\phi_{j}\left(t\right)\right)\sin\left(\phi_{j}\left(t\right)\right)\left[K_{t}\cos\left(\phi_{j}\left(t\right)\right)+K_{n}\sin\left(\phi_{j}\left(t\right)\right)\right]$$
where $K_{t}$ and $K_{n}$ denote the tangential and normal linearised cutting force coefficients, respectively. The $\phi_{j}\left(t\right)$ denotes the angular position of the *j*th tooth.

Based on the Volterra integral equation of the second type [30], the analytical expression of (6) is deduced as follows:(11)$$x\left(t\right)=e^{A\left(t-t_{st}\right)}x\left(t_{st}\right)+a_{p}\int_{t_{st}}^{t}\left\{e^{A\left(t-\xi \right)}B\left(\xi \right)\left[x\left(\xi \right)-x\left(\xi -T\right)\right]\right\}d\xi$$
where $t_{st}$ is the starting time of the computation.

Depending on whether the cutter touches the workpiece, the vibrations can be classified into free and excited vibration process [31]. If the cutter is not in contact with the workpiece, the vibrations are free vibration processes, which are equivalent to $B\left(t\right)=0$. Then, Equation (11), can be simplified as
(12)$$x\left(t\right)=e^{At_{f}}x\left(t_{st}\right)$$
where $t_{f}$ is the free vibration cycle.

If the cutter is in contact with the workpiece, the vibrations are excited vibration processes. The corresponding cycle $t_{i}$ is
(13)$$t_{i}=t_{st}+t_{f}+\left(i-1\right)h$$
where i=2m
$m=0,1,2,\cdots ,\mathrm{ceil}\left(n/2\right)-1$, where ceil (·) is a function that rounds positive numbers to plus infinity) and $h=\left(T-t_{f}\right)/n$.

By substituting Equation (13), into Equation (11), we obtain
(14)$$x\left(t_{i+1}\right)=e^{A\left(t_{i+1}-t_{i}\right)}x\left(t_{i}\right)+a_{p}\int_{t_{i}}^{t_{i+1}}\left\{e^{A\left(t_{i+1}-\xi \right)}B\left(\xi \right)\left[x\left(\xi \right)-x\left(\xi -T\right)\right]\right\}d\xi$$

Based on the composite Newton–Cotes formula introduced in Section 2, if the time interval $\left[t_{i},t_{i+1}\right]$ is divided into four equal sub-intervals, that is, $t_{i}$, $t_{i+1/4}$, $t_{i+1/2}$, $t_{i+3/4}$, and $t_{i+1}$, then the state items $x\left(t_{i+1/4}\right)$, $x\left(t_{i+1/2}\right)$, and $x\left(t_{i+3/4}\right)$ can be approximated as follows:(15)$$x\left(t_{i+1/4}\right)≃\frac{21x\left(t_{i}\right)+14x\left(t_{i+1}\right)-3x\left(t_{i+2}\right)}{32},$$
(16)$$x\left(t_{i+1/2}\right)≃\frac{3x\left(t_{i}\right)+6x\left(t_{i+1}\right)-x\left(t_{i+2}\right)}{8},$$
(17)$$x\left(t_{i+3/4}\right)≃\frac{5x\left(t_{i}\right)+30x\left(t_{i+1}\right)-3x\left(t_{i+2}\right)}{32}.$$

To simplify the process, several abbreviated expressions are used as below. Let
(18)$$x_{i}=x\left(t_{i}\right)$$
(19)$$B_{i}=B\left(t_{i}\right)$$
(20)$$\sigma \left(t,\xi ,x\left(\xi \right)\right)=a_{p}e^{A\left(t-\xi \right)}B\left(\xi \right)\left[x\left(\xi \right)-x\left(\xi -T\right)\right]$$

By combining Equations (15)–(20), $x\left(t_{i+1}\right)$ can be computed by
(21)$$\begin{array}{ll} x_{i+1}= & e^{Ah}x_{i}+\frac{h}{90}\left\{7\left[\sigma \left(t_{i+1},t_{i},x_{i}\right)+\sigma \left(t_{i+1},t_{i+1},x_{i+1}\right)\right]+12\sigma \left(t_{i+1},t_{i+1/2},\frac{3x_{i}+6x_{i+1}-x_{i+2}}{8}\right)\right.\\ & \left.+32\left[\sigma \left(t_{i+1},t_{i+1/4},\frac{21x_{i}+14x_{i+1}-3x_{i+2}}{32}\right)+\sigma \left(t_{i+1},t_{i+3/4},\frac{5x_{i}+30x_{i+1}-3x_{i+2}}{32}\right)\right]\right\} \end{array}$$

On this basis, we utilise Simpson’s rule to calculate $x_{i+2}$:(22)$$x_{i+2}=e^{2Ah}x_{i}+\frac{a_{p}h}{3}\left[e^{2Ah}B_{i}\left(x_{i}-x_{i+T}\right)+4e^{Ah}B_{i+1}\left(x_{i+1}-x_{i+1-T}\right)+B_{i+2}\left(x_{i+2}-x_{i+2-T}\right)\right].$$

By combining Equations (21) and (22), we can obtain general iterative algorithms, and the state transition equations can be written as
(23)$$F_{i}x_{i}+F_{i+1}x_{i+1}+F_{i+2}x_{i+2}=F_{i-T}x_{i-T}+F_{i+1-T}x_{i+1-T}+F_{i+2-T}x_{i+2-T}$$
(24)$$G_{i}x_{i}+G_{i+1}x_{i+1}+G_{i+2}x_{i+2}=G_{i-T}x_{i-T}+G_{i+1-T}x_{i+1-T}+G_{i+2-T}x_{i+2-T}$$
where
(25)$$F_{i}=-e^{Ah}-\frac{a_{p}h}{180}\left(14e^{Ah}B_{i}+42e^{3Ah/4}B_{i+1/4}+9e^{Ah/2}B_{i+1/2}+10e^{Ah/4}B_{i+3/4}\right)$$
(26)$$F_{i+1}=I-\frac{a_{p}h}{90}\left(7e^{Ah}B_{i+1}+14e^{3Ah/4}B_{i+1/4}+9e^{Ah/2}B_{i+1/2}+30e^{Ah/4}B_{i+3/4}\right)$$
(27)$$F_{i+2}=\frac{a_{p}h}{60}\left(2e^{3Ah/4}B_{i+1/4}+e^{Ah/2}B_{i+1/2}+2e^{Ah/4}B_{i+3/4}\right)$$
(28)$$F_{i-T}=-\frac{a_{p}h}{180}\left(14e^{Ah}B_{i}+42e^{3Ah/4}B_{i+1/4}+9e^{Ah/2}B_{i+1/2}+10e^{Ah/4}B_{i+3/4}\right)$$
(29)$$F_{i+1-T}=-\frac{a_{p}h}{90}\left(7e^{Ah}B_{i+1}+14e^{3Ah/4}B_{i+1/4}+9e^{Ah/2}B_{i+1/2}+30e^{Ah/4}B_{i+3/4}\right)$$
(30)$$F_{i+2-T}=\frac{a_{p}h}{60}\left(2e^{3Ah/4}B_{i+1/4}+e^{Ah/2}B_{i+1/2}+2e^{Ah/4}B_{i+3/4}\right)$$
(31)$$G_{i}=-e^{2Ah}-\frac{a_{p}h}{3}e^{2Ah}B_{i}$$
(32)$$G_{i+1}=-\frac{4a_{p}h}{3}e^{Ah}B_{i+1}$$
(33)$$G_{i+2}=I-\frac{a_{p}h}{3}B_{i+2}$$
(34)$$G_{i-T}=-\frac{a_{p}h}{3}e^{2Ah}B_{i}$$
(35)$$G_{i+1-T}=-\frac{4a_{p}h}{3}e^{Ah}B_{i+1}$$
(36)$$G_{i+2-T}=-\frac{a_{p}h}{3}B_{i+2}$$
where **I** is the identity matrix.

After solving the equations above, the state transition matrix is established as follows:(37)$$\left(I-P-\frac{a_{p}h}{90}Q\right)\left(\begin{array}{c} x\left(t_{0}\right)\\ x\left(t_{1}\right)\\ \vdots \\ x\left(t_{n+2}\right) \end{array}\right)=\left(R-\frac{a_{p}h}{90}Q\right)\left(\begin{array}{c} x\left(t_{0}-T\right)\\ x\left(t_{1}-T\right)\\ \vdots \\ x\left(t_{n+2}-T\right) \end{array}\right)$$
where
(38)$$P={\left(\begin{array}{ccccccc} 0 & 0 & 0 & \cdots & 0 & 0 & 0\\ e^{Ah} & 0 & 0 & \cdots & 0 & 0 & 0\\ e^{2Ah} & 0 & 0 & \cdots & 0 & 0 & 0\\ \vdots & \vdots & \vdots & \ddots & \vdots & \vdots & \vdots \\ 0 & 0 & 0 & \cdots & 0 & 0 & 0\\ 0 & 0 & 0 & \cdots & e^{Ah} & 0 & 0\\ 0 & 0 & 0 & \cdots & e^{2Ah} & 0 & 0 \end{array}\right)}_{\left(2n+2\right)\times \left(2n+2\right)}$$
(39)$$Q={\left(\begin{array}{ccccccc} 0 & 0 & 0 & \cdots & 0 & 0 & 0\\ \alpha_{0} & \alpha_{1} & \alpha_{2} & \cdots & 0 & 0 & 0\\ \beta_{0} & \beta_{1} & \beta_{2} & \cdots & 0 & 0 & 0\\ \vdots & \vdots & \vdots & \ddots & \vdots & \vdots & \vdots \\ 0 & 0 & 0 & \cdots & \beta_{i} & 0 & 0\\ 0 & 0 & 0 & \cdots & \alpha_{i} & \alpha_{i+1} & \alpha_{i+2}\\ 0 & 0 & 0 & \cdots & \beta_{i} & \beta_{i+1} & \beta_{i+2} \end{array}\right)}_{\left(2n+2\right)\times \left(2n+2\right)}$$
(40)$$R={\left(\begin{array}{ccccccc} 0 & 0 & 0 & \cdots & 0 & 0 & e^{Ah}\\ 0 & 0 & 0 & \cdots & 0 & 0 & 0\\ 0 & 0 & 0 & \cdots & 0 & 0 & 0\\ \vdots & \vdots & \vdots & \ddots & \vdots & \vdots & \vdots \\ 0 & 0 & 0 & \cdots & 0 & 0 & 0\\ 0 & 0 & 0 & \cdots & 0 & 0 & 0\\ 0 & 0 & 0 & \cdots & 0 & 0 & 0 \end{array}\right)}_{\left(2n\times 2\right)\times \left(2n+2\right)}$$
where
(41)$$\alpha_{i}=7e^{Ah}B_{i}+21e^{3Ah/4}B_{i+1/4}+\frac{9}{2}e^{Ah/2}B_{i+1/2}+5e^{Ah/4}B_{i+3/4}$$
(42)$$\alpha_{i+1}=7e^{Ah}B_{i+1}+14e^{3Ah/4}B_{i+1/4}+9e^{Ah/2}B_{i+1/2}+30e^{Ah/4}B_{i+3/4}$$
(43)$$\alpha_{i+2}=-30e^{3Ah/4}B_{i+1/4}-\frac{3}{2}e^{Ah/2}B_{i+1/2}-3e^{Ah/4}B_{i+3/4}$$
(44)$$\beta_{i}=30e^{2Ah}B_{i}$$
(45)$$\beta_{i+1}=120e^{Ah}B_{i+1}$$
(46)$$\beta_{i+2}=30B_{i+2}$$

The state transition matrix of the one tooth passing period is expressed as
(47)$$\Phi ={\left(I-P-\frac{a_{p}h}{90}Q\right)}^{-1}\left(R-\frac{a_{p}h}{90}Q\right)$$

According to the Floquet theory, if any module of the eigenvalues of the state transition matrix exceeds one, the system motion is unstable; otherwise, if all modules of the eigenvalues of the state transition matrix are less than one, the system motion is stable [32]. Therefore, the boundary curve for the stable and unstable regions in the lobe diagram can be used as a criterion to judge whether chatter occurs.

## 4. Simulation Analysis

To verify the effectiveness and superiority of the proposed method in terms ofconvergence rate, prediction accuracy, and computational efficiency, a comparison between the NCM and computer numerical control milling (CNCM) methods proposed in this paper was conducted. Herein, a single-DOF benchmark example is utilized for verification, and its mathematical model [22] is written as
(48)$$\ddot{x}\left(t\right)+2\zeta \omega_{n}\dot{x}\left(t\right)+\omega_{n}^{2}x\left(t\right)=-\frac{a_{p}h\left(t\right)}{m_{t}}\left[x\left(t\right)-x\left(t-T\right)\right]$$
where ζ denotes the damping ratio, $\omega_{n}$ denotes the natural frequency, and $m_{t}$ denotes the modal mass of the tool. The $h\left(t\right)$ denotes the specific cutting force coefficient, which is given by Equation (10). The state-space expression of Equation (48) is the same as Equation (6), where $A\left(t\right)$ and $B\left(t\right)$ satisfy
(49)$$A\left(t\right)=\left(\begin{array}{cc} 0 & I\\ -M^{-1}K & -M^{-1}C \end{array}\right)$$
(50)$$B\left(t\right)=\left(\begin{array}{cc} 0 & 0\\ -M^{-1}K_{C}\left(t\right) & 0 \end{array}\right)$$

For an equal comparison, the parameters used in this section and listed in Table 1 are the same as those in [24], and all algorithms run on the same desktop computer (AMD Ryzen 5 5600H; CPU 4.0 GHz, 16 GB).

### 4.1. Convergence Rate

The convergence rate is regarded as an important evaluation index of algorithm accuracy. It can be expressed as the difference between the approximate and exact values. Li et al. [24] proved that the local discretization error of NCM is $o\left(h^{5}\right)$. According to the discussion above, three intermediate interpolation points are considered, which makes the local discretization error of the CNCM proposed in this paper reach $o\left(h^{6}\right)$. The convergence rate can be further illustrated by a single-DOF benchmark example with the following machining conditions. The different spindle speeds and depths of cut are used: Ω = 6000 rpm, *a_p_* = 0.28 mm; Ω = 6000 rpm, *a_p_* = 0.8 mm; Ω = 6000 rpm, *a_p_* = 1 mm; Ω = 10,000 rpm, *a_p_* = 0.28 mm; Ω = 10,000 rpm, *a_p_* = 0.8 mm; Ω = 10,000 rpm, *a_p_* = 1 mm. The exact value of λ0 is calculated using SDM with n = 500. Figure 1 presents the convergence rates of the NCM and CNCM proposed in this paper. The comparison results show that the CNCM proposed in this paper obtains a higher convergence rate than the NCM. In addition, the NCM suffers from more violent oscillations than the CNCM proposed in this paper. These phenomena mentioned above can illustrate that CNCM can quickly and stably reach the target value. To further analyzethe superiority of CNCM, prediction accuracy is carried out.

### 4.2. Prediction Accuracy

The stability lobe diagrams are given in Table 2 to compare the prediction accuracies of the NCM and CNCM proposed in this paper. The time intervals of the NCM and CNCM proposed in this paper are set to 26, 40, and 56. The reference stability boundaries denoted by the black dashed line are calculated by the SDM with *n* = 500. The stability lobe diagrams are constructed over a 200 × 100 grid, and the machining parameters are set as follows: Ω∈[5000 rpm, 10,000 rpm] and *a_p_*∈ [0 mm, 5 mm]. It can be seen that the prediction accuracy of the stability lobe diagrams calculated by the CNCM proposed in this paper is higher than that calculated by the NCM.

To further verify the prediction accuracy, the arithmetic mean of the relative error (*AMRE*) and the mean squared error (*MSE*) [33] are introduced. They are defined as
(51)$$AMRE=\frac{1}{n_{e}}\sum_{i=1}^{n_{e}}\frac{\left|a_{i}-a_{i0}\right|}{a_{i0}},$$
(52)$$MSE=\frac{1}{n_{e}}\sum_{i=1}^{n_{e}}{\left(a_{i}-a_{i0}\right)}^{2},$$
where *a_i_* and *a_i_*_0_ denote the predicted and reference axial cutting depths, respectively, and *n_e_* denotes the number of discrete points. The *AMRE*s and *MSE*s are depicted in Figure 2. The *AMRE* results are given below: When the time interval is 26, the *AMRE*s of the NCM and CNCM proposed in this paper are 21.31% and 6.69%, respectively. When the time interval is 40, the *AMRE*s of the NCM and CNCM proposed in this paper are 3.88% and 0.41%, respectively. When the time interval is 56, the *AMRE*s of the NCM and CNCM proposed in this paper are 0.84% and 0.41%, respectively. The *MSE* results are given below: When the time interval is 26, the *MSE*s of the NCM and CNCM proposed in this paper are 2.46 × 10^−7^ and 9.60 × 10^−8^, respectively. When the time interval is 40, the *MSE*s of the NCM and CNCM proposed in this paper are 9.85 × 10^−9^ and 2.86 × 10^−10^, respectively. When the time interval is 56, the *MSE*s of the NCM and CNCM proposed in this paper are 1.73 × 10^−10^ and 5.28 × 10^−11^, respectively. The results above demonstrate that the proposed CNCM can achieve higher prediction accuracy than the NCM.

### 4.3. Computational Efficiency

A stability analysis method with a high convergence rate typically requires more computation time [34]. To verify that the proposed method has a relatively ideal computation efficiency, the computational times of the NCM and CNCM at the same *AMRE* and *MSE* are discussed. As shown in Figure 3, when *AMRE* = 1.75%, *n* = 43 and *t* = 43.31 s are determined by the NCM, whereas *n* = 30 and *t* = 37.10 s are determined by the CNCM proposed in this paper. It is noted that the computational time can be significantly reduced with the CNCM proposed in this paper. It can also be seen that, although the two discrete numbers are different, the prediction stability boundary obtained with the CNCM proposed in this paper is below the reference stability boundary when Ω∈ [5000 rpm, 6000 rpm]. To avoid chatter, the machining parameters selected according to the above-mentioned prediction stability boundary are more reliable. In addition, when *MSE* = 1.12 × 10^−10^, the discrete numbers and the computational time, as well as the prediction stability boundaries, are shown in Figure 4. The discrete number obtained with the NCM is greater than that obtained with the CNCM proposed in this paper. However, the computational efficiency of the NCM is slightly lower than that of the CNCM proposed in this paper.

## 5. Experimental Verification

To further verify the effectiveness and superiority of the proposed CNCM, milling experiments were carried out with a feed per tooth of 0.06 mm, Ω∈ [3000 rpm, 8000 rpm] and *a_p_*∈ [0 mm, 4 mm]. The milling system was a benchmark system with two DOFs. The workpiece material was aluminum 7075. Before milling experiments, the frequency response function (FRF) of tool should be obtained first. Thus, the hammer experiment is carried out. The measuring instruments include the Kistler 9724A5000 force hammer, Endvco 2256B-10 acceleration sensor, ATI omega-160 force sensor, DEWEsoft SIRIUSI-8xACC data acquisition system and DEWEsoft data analysis system, as shown in Figure 5. The physical and machining parameters of the Harmer experiment are listed in Table 3. A stability lobe diagram of the proposed CNCM is depicted in Figure 6. It can be noted that 20 points are selected from the stability lobe diagram as the test points.

The amplitude spectrum of the milling force signal can show many details in the frequency domain, which can be used to verify the prediction accuracy of the SLD presented in Figure 6. Ten critical points are selected as the milling parameters, and the milling force is measured by an ATI force sensor with a sample rate of 8000. When the milling process is stable, there is only one tooth passing frequency in the spectrum. However, when chatter occurs, there will be a series of disorders in the frequency spectrum. As shown in Figure 7, the milling system has the tooth passing frequency and its harmonic frequency (red value) during stable cutting situations corresponding to A1 (3500 rpm, 0.5 mm), B2 (4500 rpm, 1 mm), C1 (5500 rpm, 0.5 mm), D3 (6500 rpm, 1.5 mm), D4 (6500 rpm, 2 mm), and E1 (7500 rpm, 0.5 mm). It also has chatter frequency (blue value) during unstable cutting situations, corresponding to A2 (3500 rpm, 1 mm), B3 (4500 rpm, 1.5 mm), C2 (5500 rpm, 1 mm), and E2 (7500 rpm, 1 mm). It can be concluded that select milling parameters above the boundary will chatter, while those below the boundary can stabilize the milling process.

Take A1–A4 as examples. A1: Ω = 3500 rpm and *a_p_* = 0.5 mm. A2: Ω = 3500 rpm and *a_p_* = 1 mm. A3: Ω = 3500 rpm and *a_p_* = 1.5 mm. A4: Ω = 3500 rpm and *a_p_* = 2 mm. The machined surfaces corresponding to A1–A4 are shown in Figure 8. When chatter occurs, the quality of the machined surface is poor. It can be seen that the chatter becomes increasingly obvious as the depth of the cut increases. The machined surface roughness values are listed in Table 4. Under the conditions of A1–A4, the average values of the machined surface roughness are 1.358, 2.392, 2.462, and 2.910 μm, respectively. It is obvious that the chatter significantly decreases the machined surface quality. The machined surface roughness under the conditions of B1–B4, C1–C4, D1–D4, and E1–E4 is also listed in Table 4. It can be observed that, under the conditions of A1, B1, B2, C1, D1–D4, and E1, the average values of the machined surface roughness are 1.358, 1.181, 1.501, 1.363, 1.312, 1.514, 1.655, 1.937, and 1.305 μm, respectively. The results show that the cutting parameters below the stability boundary can avoid chatter as much as possible, thereby ensuring the quality of the machined surface.

## 6. Conclusions

This study focuses on the convergence rate and prediction accuracy of the milling stability based on the composited Newton–Cotes formula. For example, when the discrete number is 40, the convergence rate and prediction accuracy improve about 4.1 and 9.4 times, respectively, while computational efficiency is similar. The reason is that because the proposed method considers three intermediate interpolation points, the local discrete error of the proposed method is smaller than that of the comparison method, which ensures the prediction accuracy of the proposed method. The state transition matrix of the proposed method is divided into three submatrices, which can avoid double calculation and matrix multiplication. The simulation results of the milling model with a single DOF validate that the proposed method has a high convergence rate and prediction accuracy while ensuring computational efficiency. The experimental results of the milling model with two DOFs validate that the machining parameters below the prediction stability boundary obtained with the proposed method can avoid chatter as much as possible while ensuring the machined surface quality.

## Figures and Tables

**Figure 1 micromachines-14-01304-f001:**
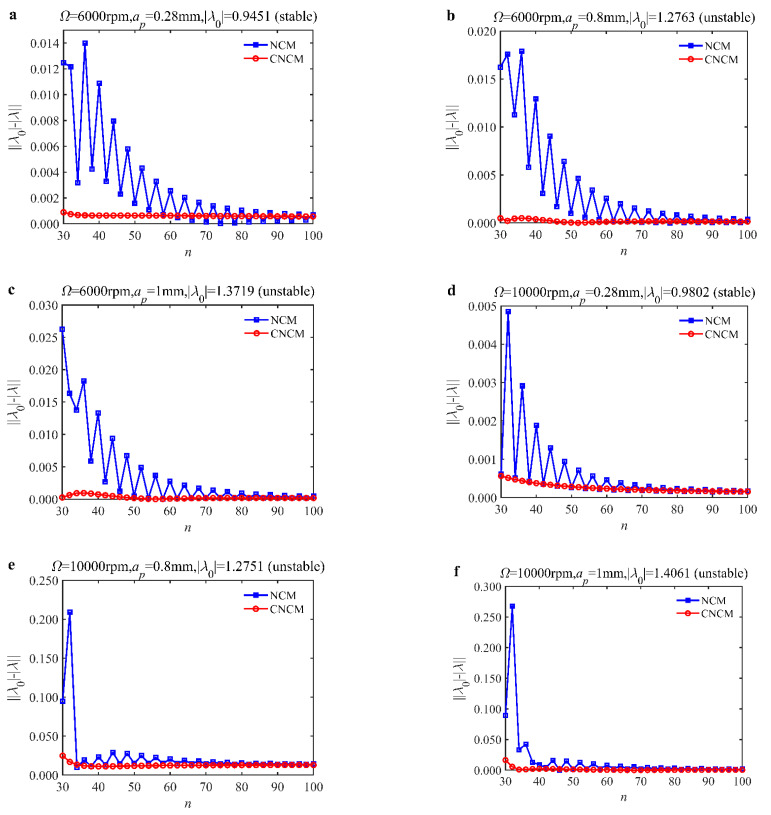
Convergence rate comparisons between the NCM and the CNCM proposed in this paper. (**a**) Ω = 6000 rpm, *a_p_* = 0.28 mm, and $\left|\lambda_{0}\right|=0.9451$ (stable); (**b**) Ω = 6000 rpm, *a_p_* = 0.8 mm, and $\left|\lambda_{0}\right|=1.2763$ (unstable); (**c**) Ω = 6000 rpm, *a_p_* = 1 mm, and $\left|\lambda_{0}\right|=1.3719$ (unstable); (**d**) Ω = 10,000 rpm, *a_p_* = 0.28 mm, and $\left|\lambda_{0}\right|=0.9802$ (stable); (**e**) Ω = 10,000 rpm, *a_p_* = 0.8 mm, and $\left|\lambda_{0}\right|=1.2751$ (unstable); (**f**) Ω = 10,000 rpm, *a_p_* = 1 mm, and $\left|\lambda_{0}\right|=1.4061$ (unstable).

**Figure 2 micromachines-14-01304-f002:**
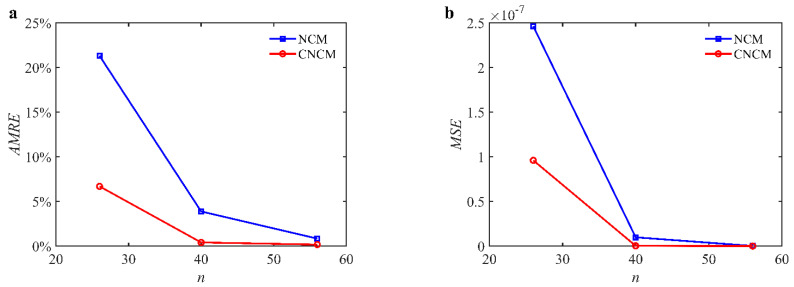
*AMRE* and *MSE* comparisons between the NCM and the CNCM proposed in this paper. (**a**) *AMRE* comparisons when *n* = 26, *n* = 40 and *n* = 56; (**b**) *MSE* comparisons when *n* = 26, *n* = 40 and *n* = 56.

**Figure 3 micromachines-14-01304-f003:**
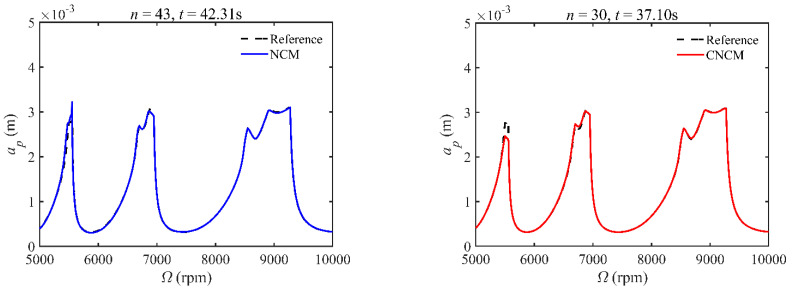
Computational time comparisons between the NCM and the CNCM proposed in this paper when *AMRE* = 1.75%.

**Figure 4 micromachines-14-01304-f004:**
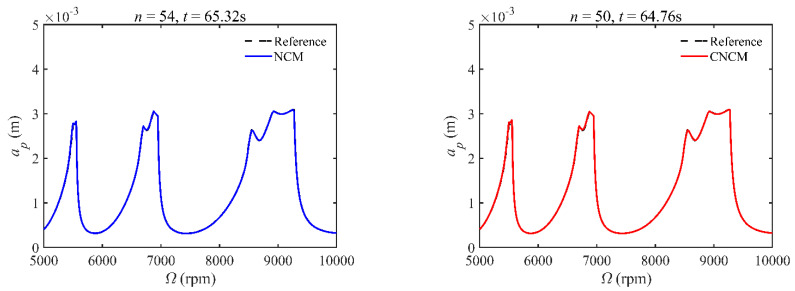
Computational time comparisons between the NCM and the CNCM proposed in this paper when *MSE* = 1.12 × 10^−10^.

**Figure 5 micromachines-14-01304-f005:**
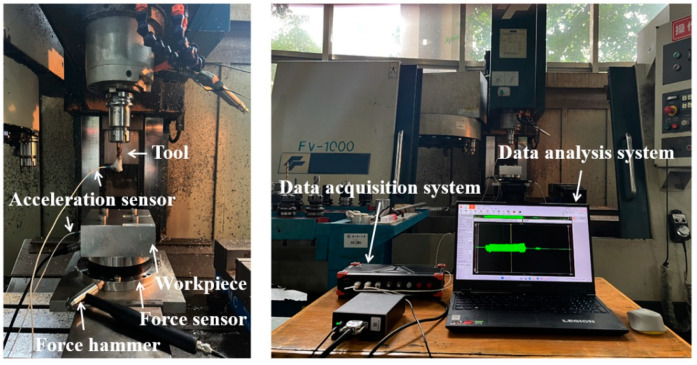
Experiments platform, measuring instruments and calibration process.

**Figure 6 micromachines-14-01304-f006:**
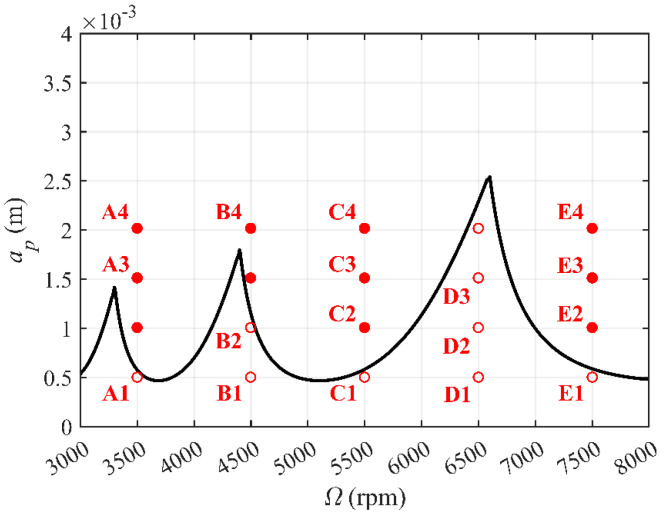
Predicted stability lobe diagram with the CNCM proposed in this paper, where hollow and filled circles denote stable and unstable cutting situations, respectively. A1: Ω = 3500 rpm, *a_p_* = 0.5 mm; A2: Ω = 3500 rpm, *a_p_* = 1 mm; A3: Ω = 3500 rpm, *a_p_* = 1.5 mm; A4: Ω = 3500 rpm, *a_p_* = 2 mm. B1: Ω = 4500 rpm, *a_p_* = 0.5 mm; B2: Ω = 4500 rpm, *a_p_* = 1 mm; B3: Ω = 4500 rpm, *a_p_* = 1.5 mm; B4: Ω = 4500 rpm, *a_p_* = 2 mm. C1: Ω = 5500 rpm, *a_p_* = 0.5 mm; C2: Ω = 5500 rpm, *a_p_* = 1 mm; C3: Ω = 5500 rpm, *a_p_* = 1.5 mm; C4: Ω = 5500 rpm, *a_p_* = 2 mm. D1: Ω = 6500 rpm, *a_p_* = 0.5 mm; D2: Ω = 6500 rpm, *a_p_* = 1 mm; D3: Ω = 6500 rpm, *a_p_* = 1.5 mm; D4: Ω = 6500 rpm, *a_p_* = 2 mm. E1: Ω = 7500 rpm, *a_p_* = 0.5 mm; E2: Ω = 7500 rpm, *a_p_* = 1 mm; E3: Ω = 7500 rpm, *a_p_* = 1.5 mm; E4: Ω = 7500 rpm, *a_p_* = 2 mm.

**Figure 7 micromachines-14-01304-f007:**
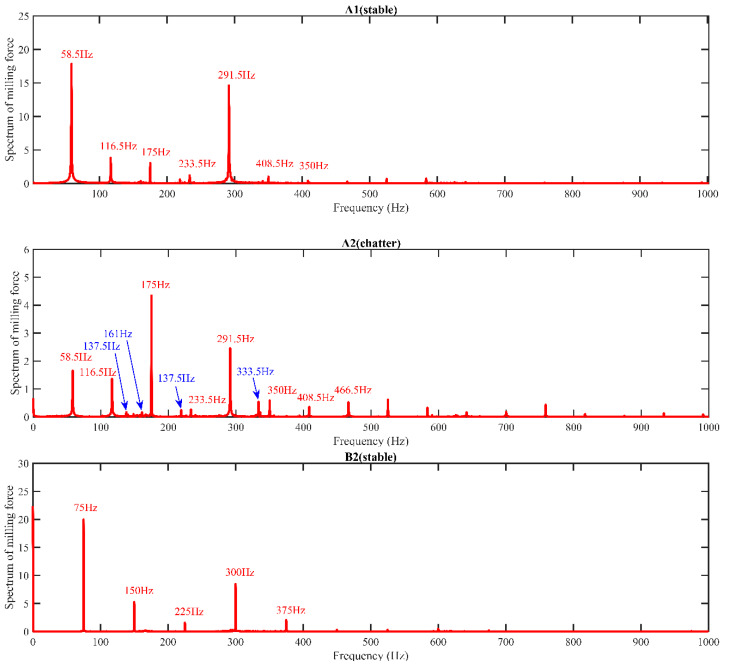
The amplitude spectrum of measured points.

**Figure 8 micromachines-14-01304-f008:**
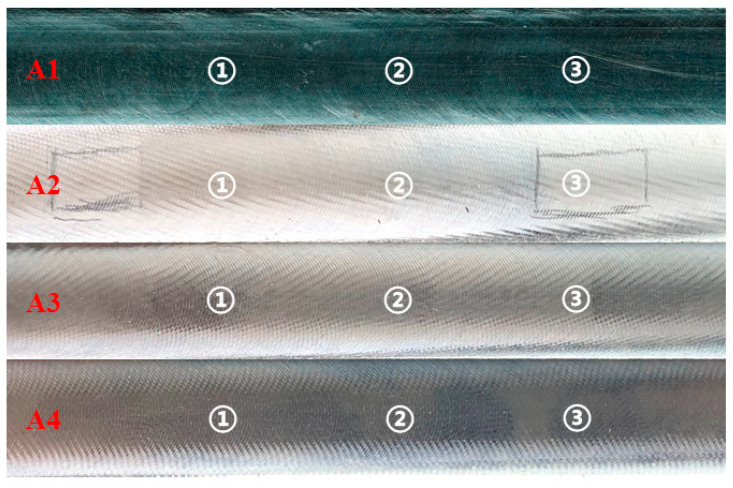
Machined surface of Aluminium 7075 with the cutting parameters corresponding to A1–A4, where 1, 2, and 3 in white denote three different sampling points.

**Table 1 micromachines-14-01304-t001:** Physical and machining parameters of the benchmark example with a single DOF [24].

Parameters	Symbols	Values
Modal mass (kg)	*m_t_*	0.03993
Damping ratio	*ζ*	0.011
Natural frequency (Hz)	*ω_n_*	922 × 2π
Tangential linearized cutting force coefficient (N/m)	*K_t_*	6 × 10^8^
Normal linearized cutting force coefficient (N/m)	*K_n_*	2 × 10^8^
Number of tool teeth	*N_t_*	2
Radial immersion ratio	*a*/*D*	1

**Table 2 micromachines-14-01304-t002:** Stability lobe diagrams with the NCM and CNCM proposed in this paper.

*n*	NCM	CNCM
26	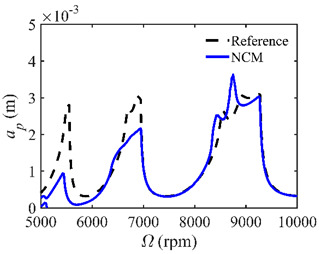	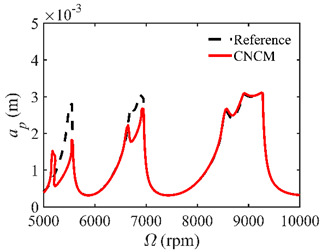
40	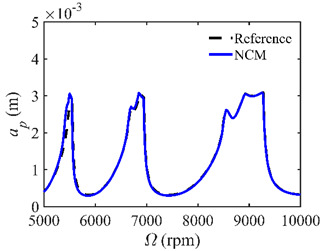	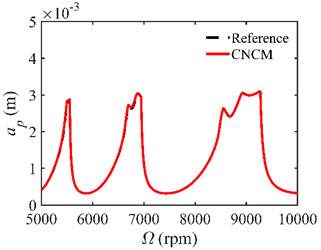
56	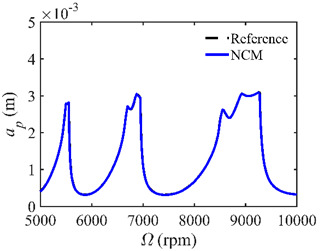	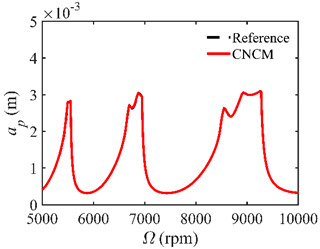

**Table 3 micromachines-14-01304-t003:** Physical and machining parameters of the Harmer experiment with two DOFs.

Parameters	Symbols	Values
Modal mass (kg)	*m_t_*	0.274
Damping ratio	*ζ*	0.0338
Natural frequency (Hz)	*ω_n_*	876 × 2π
Tangential linearized cutting force coefficient (N/m)	*K_t_*	5.5 × 10^8^
Normal linearized cutting force coefficient (N/m)	*K_n_*	3.75 × 10^8^
Number of tool teeth	*N_t_*	4
Tool diameter (m)	*D*	0.012
Radio immersion ratio	*a*/*D*	1

**Table 4 micromachines-14-01304-t004:** Machined surface roughness of Aluminium 7075 with the machining parameters corresponding to A1–A4, B1–B4, C1–C4, D1–D4, and E1–E5.

Serial Numbers	Surface Roughness of the Measured Points (μm)			
	1	2	3	Average Values
A1	1.490	1.647	1.738	1.625
A2	2.997	2.053	2.127	2.392
A3	2.879	2.135	2.371	2.462
A4	2.967	2.817	2.947	2.910
B1	1.068	1.190	1.286	1.181
B2	1.496	1.447	1.561	1.501
B3	2.287	2.332	2.317	2.312
B4	2.242	2.294	2.143	2.226
C1	1.089	1.594	1.407	1.363
C2	2.335	2.486	2.358	2.393
C3	2.423	2.506	2.712	2.547
C4	2.673	2.685	2.837	2.732
D1	1.265	1.330	1.340	1.312
D2	1.563	1.453	1.526	1.514
D3	1.744	1.475	1.745	1.655
D4	1.997	1.945	1.868	1.937
E1	1.374	1.269	1.273	1.305
E2	3.286	2.881	3.066	3.078
E3	3.293	3.792	2.775	3.287
E4	3.609	3.360	3.379	3.449

## Data Availability

The data presented in this study are available on request from the corresponding author.

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
