# Peer review of "Milling Stability Prediction: A New Approach Based on a Composited Newton–Cotes Formula"

_micromachines, 2023, doi:10.3390/mi14071304_

Round 1

Reviewer 1 Report

 In this paper,a numerical method of predicting milling stability considering a regenerative chatter is proposed, and many experiments are carried out, the reliability of the prediction model proposed is validated. This paper is rich in content, but it needs a major revision according to the following comments:

1. There have been many papers on stability prediction, and the superiority of your algorithm has not been explained in detail in the text

2. The correlation between your algorithm and experimental results are described in Fig. 7 and Fig. 8, in addition, the criteria for determining whether chatter occurs or not should be analyzed.

3. The analysis of experimental results is not deep enough.

4. Some grammatical and word errors need to be corrected.

No

Author Response

In this paper, a numerical method of predicting milling stability considering a regenerative chatter is proposed, and many experiments are carried out, the reliability of the prediction model proposed is validated. This paper is rich in content, but it needs a major revision according to the following comments.

Q1. There have been many papers on stability prediction, and the superiority of your algorithm has not been explained in detail in the text.

Response: Thank you for the reminder. The superiority of the proposed milling dynamic algorithm has been discussed in Section 4. We also further analysis the advantages of our algorithm. Please look at Section 4.1, 4.2 and 4.3.

Q2. The correlation between your algorithm and experimental results are described in Fig. 7 and Fig. 8, in addition, the criteria for determining whether chatter occurs or not should be analyzed.

Response: We have added the criteria for determining whether chatter occurs or not in section 5. First of all, milling chatter can be detected in frequency domain. We use the milling force during the process as the import data. Then utilizing the Fast Fourier Transform algorithm to convert time domain data into frequency domain. Then look at the spectrum diagram. If there are tooth passing frequency and its harmonic frequency in the diagram, system is stable. If there exist some strange frequency spectrum, system is unstable. Secondly, the surface quality of machined workpiece can also reflect the system is stable or not. So, we also use the roughness as the criterion.   

Q3. The analysis of experimental results is not deep enough.

Response: Thank you for the reminder. We have updated the analysis of experimental results.

 Q4. Some grammatical and word errors need to be corrected.

Response: Thank you for the reminder. We have updated the text.

Reviewer 2 Report

[1] What are the assumptions for the tool and the milling process kinematics in the presented method?

[2] Please give a figure to describe the milling process.

[3] Below Eq.(5), the M C K should be bold

Author Response

Q1. What are the assumptions for the tool and the milling process kinematics in the presented method?

Response: In this work, it is assumed that the workpiece is rigid, that the milling cutters have two DOFs and that there is no tool run-out situation in milling process. The tool helix angle is not considered either. We have added the assumptions in Section 3.

 Q2. Please give a figure to describe the milling process.

Response: Thank you for the reminder. We have added the figure in Section 5.

 Q3. Below Eq.(5), the M C K should be bold.

Response: Thank you for the reminder. We have bolded M C K.

Reviewer 3 Report

This study focuses on the convergence rate and prediction accuracy of the milling stability based on the composited Newton-Cotes formula.The method is validated through simulation and experiments.There are the following issues:

1.What is the reason for dividing [xk, xk+1] into four intervals in Section 2.1?

2. How much has the Rate of convergence and accuracy improved? It is better to explain in the conclusion.

3.References need to be updated.

4. The clarity of some images in sections 4 and 5 needs to be improved.

Author Response

This study focuses on the convergence rate and prediction accuracy of the milling stability based on the composited Newton-Cotes formula. The method is validated through simulation and experiments. There are the following issues:

Q1. What is the reason for dividing [Xk,, Xk+1] into four intervals in Section 2.1?

Response: To improve the integration accuracy, mid-points of the interval should be considered. So we dividing the interval into four equal parts. And we added the reason above in Section 2.

 Q2. How much has the Rate of convergence and accuracy improved? It is better to explain in the conclusion.

Response: For example, when discrete number is 40, the convergence rate and prediction accuracy improve about 4.1 and 9.4 times respectively while computational efficiency is similar. We have added it in conclusion.

 Q3. References need to be updated.

Response: Thank you for the reminder. We have updated some references. However, some references are classic milling stability analysis method which are mentioned in introduction. So, we can’t change these kinds of references.

 Q4. The clarity of some images in sections 4 and 5 needs to be improved.

Response: Thank you for the reminder. We have updated these images.

Round 2

Reviewer 3 Report

The modifications have been made as required.